# Test–Retest Reliability and Internal Consistency of a Newly Developed Questionnaire to Assess Explanatory Variables of 24-h Movement Behaviors in Adults

**DOI:** 10.3390/ijerph20054407

**Published:** 2023-03-01

**Authors:** Iris Willems, Vera Verbestel, Patrick Calders, Bruno Lapauw, Marieke De Craemer

**Affiliations:** 1Department of Rehabilitation Sciences, Ghent University, 9000 Ghent, Belgium; 2Research Foundation Flanders, 1000 Brussels, Belgium; 3Department of Public Health and Primary Care, Ghent University, 9000 Ghent, Belgium; 4Department of Internal Medicine and Pediatrics, Ghent University, 9000 Ghent, Belgium; 5Department of Endocrinology, Ghent University Hospital, 9000 Ghent, Belgium

**Keywords:** 24-h movement behaviors, explanatory variables, questionnaire, socio-ecological model, test–retest reliability, internal consistency

## Abstract

A questionnaire on explanatory variables for each behavior of the 24-h movement behaviors (i.e., physical activity, sedentary behavior, sleep) was developed based on three levels of the socio-ecological model, i.e., the intrapersonal level, interpersonal level and the physical environmental level. Within these levels, different constructs were questioned, i.e., autonomous motivation, attitude, facilitators, internal behavioral control, self-efficacy, barriers, subjective norm, social modeling, social support, home environment, neighborhood, and work environment. The questionnaire was tested for test–retest reliability (i.e., intraclass correlation (ICC)) for each item and internal consistency for each construct (i.e., Cronbach’s Alpha Coefficient) among a group of 35 healthy adults with a mean age of 42.9 (±16.1) years. The total questionnaire contained 266 items, consisting of 14 items on general information, 70 items on physical activity, 102 items on sedentary behavior, 45 items on sleep and 35 items on the physical environment. Seventy-one percent of the explanatory items showed moderate to excellent reliability (ICC between 0.50 and 0.90) and a majority of constructs had a good homogeneity among items (Cronbach’s Alpha Coefficient ≥ 0.70). This newly developed and comprehensive questionnaire might be used as a tool to understand adults’ 24-h movement behaviors.

## 1. Introduction

Healthy lifestyle behaviors (e.g., being physically active, interrupting sitting periods, optimal sleep pattern) have proven beneficial health effects in the prevention of numerous medical conditions such as diabetes, obesity, and cardiovascular diseases [1,2,3,4]. For a long time, these beneficial health effects have been investigated in health promotion research by focusing on one lifestyle behavior in isolation, e.g., moderate to vigorous physical activity (MVPA) [4,5]. However, a shift in research emphasizes the importance of considering all movement behaviors throughout a day, including PA, SB and sleep, as they are interconnected and mutually exclusive parts of a 24-h day [6]. This 24-h movement behavior paradigm recognizes that any change in time in one behavior inevitably leads to a change in time in one of the other behaviors [6,7,8]. While PA and SB are represented by their duration, achieving an optimal sleep pattern involves more than just duration, such as sleep timing and consistency [9].

Research on this 24-h movement behavior paradigm is rapidly expanding. A systematic review by Janssen and colleagues (2020) revealed associations between less optimal 24-h movement behavior compositions (i.e., high SB levels, low PA levels, non-optimal sleep duration) and all-cause mortality, cardiometabolic risk factors and adiposity among a general adult population [8]. Additionally, studies exploring associations between 24-h compositions and cardiometabolic health among healthy and clinical populations (e.g., diabetes) showed beneficial health effects when replacing small amounts of time (e.g., 10 min) from SB into more PA while preserving sleep duration. [10,11].

Therefore, a behavior change intervention to promote optimal 24-h movement behavior compositions holds great potential to improve health of general as well as clinical populations [7,8]. In order to obtain effective behavior change, it is fundamental to investigate all factors that determine and explain these behaviors (e.g., self-efficacy, attitude) [12]. An accurate assessment of these factors, also known as explanatory variables, allows for the development of tailored interventions targeting 24-h movement behaviors [12,13]. Questionnaires on explanatory variables of lifestyle behaviors in isolation already exist (e.g., Determinants of Physical Activity Questionnaire (DPAQ), the Sedentary Behavior Change Questionnaire). Until now, there has been no questionnaire conceptualized from a 24-h movement behavior perspective, including comparable explanatory variables for PA, SB, and complying with an optimal sleeping pattern [12,14,15]. Moreover, these existing questionnaires often lack reliability and do not incorporate a theoretical behavior change framework (e.g., Theory of Planned Behavior (TPB), Self-Determination Theory (SDT)) [14]. The main advantage of incorporating a theoretical framework is the potential to use this framework as an explorative as well as an evaluative tool for the development of health promotion interventions [16,17,18,19]. Additionally, incorporating a behavioral change framework might be more effective in promoting behavior change compared to approaches lacking a theoretical foundation [16].

Theoretical frameworks that have been used in behavior change interventions to promote PA, SB and/or sleep are mainly focused on psychological factors centered within the person (e.g., attitude, self-efficacy) [20,21,22,23]. Nevertheless, intrapersonal characteristics of behavior include only one level, whereas a socio-ecological model broadens this perspective to a multilevel analysis of behavior, including physical and socio-cultural surroundings [24,25]. An example of such a multilevel analysis model of behavior is the socio-ecological model of Bronfenbrenner (1977), consisting of four levels [26]. The intrapersonal level includes demographic characteristics as well as other personal psychological factors such as beliefs, barriers, attitudes [26]. The interpersonal level collects information on the social environment such as support of family and friends [26]. The physical environmental level explores the perceived physical environment of a person [26]. The fourth level is the policy level, which includes all laws and rules on a certain behavior [26]. Additionally, Sallis and colleagues (2006) added an extra dimension of different active living domains in which it is possible to be physically active, i.e., household, leisure time, transport and work [25]. Furthermore, existing research identified associations between socio-ecological levels and lifestyle behaviors. Therefore, combining personal and environmental characteristics might be key to gain a broader perspective of explanatory variables of adults’ 24-h movement behaviors [27,28,29].

In summary, literature dealing with the 24-h movement behavior paradigm is rapidly growing and promotes the interrelatedness between daily behaviors which brings new challenges for health promotion research. Therefore, it is valuable to develop a questionnaire to assess the underlying characteristics of the overall 24-h day focusing on PA, SB and complying with and optimal sleeping pattern. The aim of this study was to investigate the test–retest reliability and internal consistency of a newly developed questionnaire on explanatory variables of 24-h movement behaviors among adults, based on a socio-ecological model.

## 2. Materials and Methods

### 2.1. Participants

A sample size of 40 adults with a minimum age of 18 years old was recruited in Flanders, Belgium [30,31,32]. The sample size was calculated for test–retest reliability to detect at least an intraclass correlation coefficient (ICC) of 0.5 (cut-off for moderate reliability) and Cronbach’s Alpha Coefficient (α) of 0.7 (cut-off for sufficient homogeneity) [30,31,32]. A minimum of 30 participants for ICC and a minimum of 24 participants for Cronbach’s Alpha Coefficient was recommended: (1) ICC: observations = 2, R_0_ = 0, min. ICC = 0.5, power = 90%, alpha = 0.05; (2) Cronbach’s Alpha Coefficient: max. items/construct = 20, R_0_ = 0, min. α =0.7, power = 90%, alpha = 0.05. Due to the comprehensiveness of the questionnaire, a drop-out rate of 25 percent was included, which created a total sample size of 40 adults [30,31,32]. Participants were included when meeting the following criteria: (1) minimum age of 18 years; (2) working for at least 50 percent; (3) not having physical (e.g., amputations, paralysis)/cognitive (e.g., dementia)/medical (e.g., heart failure, chronical obstructive pulmonary diseases) conditions that affect daily functioning. By including participants working for at least 50 percent, the working adult population is covered and full-time students and retired adults were automatically excluded.

Participants were recruited by using convenience sampling within the researchers’ network. The study was approved by the Ethics Committee of Ghent University (BC-08622). Prior to the start of the study, informed consent was obtained, which was explained to and signed by all participants.

### 2.2. Questionnaire Development

The questionnaire was developed based on the socio-ecological model of Bronfenbrenner (1977) and the active living domains of Sallis et al. (2006) [25,26]. For each 24-h movement behavior, factors within three out of the four levels of the socio-ecological model were assessed, i.e., the intrapersonal, interpersonal and physical environmental level [33]. Questions on PA contained both questions on light PA (LPA) and MVPA. Questions on SB took into account periods of long and uninterrupted SB as well as breaks in SB. Sleep was questioned as the compliance with an optimal sleeping pattern which was defined as a sleep duration ranging from 7 to 9 h and consistent wake-up and go to bed times [7]. Within each of these behaviors, different constructs were questioned, e.g., attitude, social modeling. Items within these constructs were evaluated on a 5-point Likert scale (i.e., (1) strongly disagree–strongly agree, (2) never–always, or (3) less time–a lot of time), except for two constructs—attitude and electronic devices at home. Answers on the attitude construct scale were formulated on a slider Visual Analog Scale (0–100) with five different options: annoying–nice; frustrating–satisfying; unhealthy–healthy; unimportant–important; difficult–easy. The number of electronic devices at home was quantified. Figure 1 provides an overview of the different levels of the socio-ecological model accompanied with an example of an item from the questionnaire.

### 2.3. Intrapersonal Level

The following sociodemographic variables were examined as part of the intrapersonal level: age, sex, family situation, children, neighborhood, country of birth, native language, educational level, educational level partner, profession, profession partner, net family income per month, smoking, and medication intake [34]. The combination of the net family income per month, educational level (yourself and partner) and profession (yourself and partner) provide an estimation of the socio-economic status of the participant. Other explanatory variables within the intrapersonal level are based on the integrated behavioral change (IBC) model [35]. The IBC model combines psychological factors from different behavior change theories including the TPB, SDT, Dual System Theory, and Social Cognitive Theory (SCT)) [35]. This resulted in the following psychological constructs being included in the questionnaire: autonomous motivation (Theory: SDT), attitude (Theory: TPB, SCT), internal behavioral control such as habits, routines (Theory: Dual System Theory), self-efficacy/perceived behavioral control (Theory: SCT, TPB, SDT) and external behavioral control such as barriers and facilitators (Theory: Dual System Theory) [35].

#### 2.3.1. Interpersonal Level

The interpersonal level represents the social environment and contains three constructs, i.e., subjective norm (Theory: TPB), social modeling (Theory: SCT) and social support (Theory: SDT) [25,26].

#### 2.3.2. Physical Environmental Level

The physical environmental factors reflect participants’ perceived environment and are structured within four constructs: the sleep environment within the home environment, electronic devices within the home environment, the neighborhood and the work environment [25,26].

### 2.4. Procedure

This questionnaire was built in REDCap, which is a secure, web-based software platform for data collection and management developed by Vanderbilt University (Nashville, TN, USA) [36]. This electronic data capture tool is hosted by the Health Innovation and Research Institute of Ghent University Hospital [36]. A digital version of the questionnaire was completed online by the participants. All questionnaires were completed in Dutch. The participants had to fill in the questionnaire twice, once at baseline (Timepoint T1; test) and once 14 days later (Timepoint T2; retest), which is a recommended time frame to assess test–retest reliability [37,38]. Two days prior to T2, a reminder was sent to fill in the questionnaire for the second time on the 14th day.

### 2.5. Statistical Analyses

All sample characteristics were categorical variables which are expressed as a percentage of the total sample size, except age and the time interval between T1 and T2, which is expressed as a mean with standard deviation. The test–retest reliability was examined calculating the ICC and the respective 95% confidence intervals. The ICC and their 95% confidence intervals were calculated based on a single measurement, absolute agreement, and two-way mixed effects [39]. An ICC higher than 0.90 represented excellent reliability, an ICC between 0.75 and 0.90 represented a good reliability, an ICC between 0.50 and 0.75 represented a moderate reliability, and an ICC lower than 0.5 represented a poor reliability [39]. The internal consistency between items within constructs was assessed by using the Cronbach’s Alpha Coefficient. A Cronbach’s Alpha Coefficient higher than 0.70 indicated sufficient homogeneity among items [40]. If the Cronbach’s Alpha Coefficient of a construct was below 0.70, the homogeneity among items was considered insufficient. Consequently, an evaluation of deleting specific items within the construct was done to check the homogeneity again [40]. If the homogeneity could not be improved, it was recommended to interpret every item separately with exclusion of items identified as poor reliable. All statistical tests were calculated using SPSS statistical package version 27 [41].

## 3. Results

### 3.1. Sample Characteristics

Table 1 gives an overview of the sample characteristics. Of the 40 participants who consented to participate, everyone completed the questionnaire at T1 and a total of 35 participants filled in the questionnaire at both time points. The reason for drop out was lack of time to fill in the questionnaire at T2. The 35 included participants had a mean age of 42.94 years (±16.07), and 60 percent were women. All participants were native Dutch speakers. The socio-economic status of the participants was high, as most participants had a high educational level (diploma higher then secondary school), a net family income of >2000 EUR/month, and were employed. The average time interval between T1 and T2 was 15.82 days (±2.00). The average response time to fill in the online questionnaire was subjectively reported as ranging from 30 min to 50 min.

### 3.2. Test–Retest Reliability

The total questionnaire contains 266 items, consisting of 14 items on general information, 70 items on physical activity, 102 items on sedentary behavior, 45 items on sleep and 35 items on the physical environment. Table 2 provides a detailed description of the ICC-range per construct in the questionnaire. Appendix A provides the test–retest reliability of each questionnaire item separately, i.e., ICC, lower bound and upper bound (see Appendix A).

All the sociodemographic items showed good to excellent reliability, i.e., 14 items (100.00%). Of the 70 items regarding PA, 40 items (57.14%) showed a moderate to good test–retest reliability, and 30 items (42.86%) showed poor reliability. Within the intrapersonal level, the number of items with moderate to good reliability were one item within the autonomous motivation construct (50.00%), three items within the attitude construct (30.00%), three items within the facilitators construct (21.43%), one item within the internal behavioral control construct (50.00%), eight items within the self-efficacy construct (80.00%), and 11 items within the barriers construct (68.75%). Within the interpersonal level, all three constructs showed a moderate to good reliability for most of their items, i.e., subjective norm (four items, 80.00%), social modeling (five items, 100.00%), social support (four items, 66.66%).

Out of the 102 items regarding SB, 71 (69.61%) items had a good to excellent test–retest reliability and 31 items (30.39%) showed poor reliability. Within the intrapersonal level, the number of items with moderate to good reliability were one item within the autonomous motivation construct (50.00%), three items within the construct on attitude regarding a long sitting period (60.00%), two items within the attitude regarding interrupting sitting construct (40.00%), seven items within the facilitators construct (87.50%), three items within the internal behavioral control construct (60.00%), nine items within the self-efficacy construct (52.94%), and nine items within barriers construct (56.25%). In addition, each of these overall constructs was divided into subconstructs based on the active living domains. See Table 2 for the reliability levels of each of these subconstructs.

Within the interpersonal level, all three constructs showed a moderate to good reliability for a majority of items, i.e., subjective norm (eight items, 66.67%), social modeling (18 items, 90.00%), social support (11 items, 91.67%). Again, each of these overall constructs was divided into subconstructs based on the active living domains. See Table 2 for the ICC and internal consistency of these subconstructs.

Of the forty-five items on sleep, 30 items (66.67%) were classified as moderate to good reliable and 15 items (33.33%) as poor reliable. Within the intrapersonal level, the items with moderate to good reliability corresponded to five items within the attitude construct (50.00%), three items within the facilitators construct (50.00%), two items within the internal behavioral control construct (100.00%), four items within the self-efficacy construct (66.67%), and 10 items within barriers construct (83.33%). All items within autonomous motivation had a poor reliability (100.00%). Within the interpersonal level, all three constructs showed a moderate to good reliability for most of their items, i.e., subjective norm (two items, 100.00%), social modeling (three items, 100.00%), and social support (one item, 50.00%).

Last, the physical environmental level contains four constructs. All items within the electronic devices within the home environment showed a moderate to good reliability (10 items, 100.00%). The sleep environment within the home environment contained six items with a moderate to good reliability (85.71%). The construct neighborhood consisted of 12 items with a moderate to excellent reliability (92.31%) and one item with a poor reliability (7.69%). Finally, all work environment items (five items, 100.00%) showed good to excellent reliability.

### 3.3. Internal Consistency

Table 2 represents all Cronbach’s Alpha Coefficients per construct. All PA constructs, except for autonomous motivation (α = 0.585), social modeling (α = 0.561) and social support (α = 0.623), showed sufficient homogeneity among items (α > 0.700). All SB constructs, except for attitude regarding long sitting periods (α = 0.695), internal behavioral control (α = 0.389), self-efficacy regarding sitting during household tasks (α = 0.533), subjective norm regarding passive transport (α = 0.638), and social support regarding sitting during household tasks (α = 0.645), demonstrated a sufficient homogeneity among items (α > 0.700). Additionally, all sleep constructs, except for social support (α = 0.687), resulted in a sufficient homogeneity among items (α > 0.700). Last, the neighborhood construct and the work environment construct showed a sufficient homogeneity among items (α > 0.700). The sleep environment and electronic devices within the home environment resulted in low Cronbach’s Alpha Coefficients (α = 0.526, α = 0.664). No improvement in homogeneity among items was achieved by deleting items within any of the above-mentioned constructs.

## 4. Discussion

The main aim of this study was to assess the test–retest reliability and the internal consistency of a questionnaire on explanatory variables of 24-h movement behaviors among adults. Overall, this study showed a moderate to excellent test–retest reliability for items belonging to different constructs and a good internal consistency among these constructs.

Seventy-one percent of all explanatory items showed a moderate to excellent reliability (188 out of 266 items). However, 29 percent of the items had a poor reliability. This could possibly be explained by biases linked with test–retest reliability research, i.e., “recall bias” and the “question behavior effect”. Recall bias, also known as response bias, refers to various conditions that lead to participants inaccurately responding to questions [38]. To overcome this bias, the time period of test–retest reliability studies should be optimal [37,38]. When this time period is too short, the respondents will remember their answers to questions [37,38]. When the period is too long, participants’ behavior may have changed over time. However, the time interval in this study, i.e., 14 days, is generally considered to be adequate [37,38]. Additionally, it is possible that participants focus more on their lifestyle behaviors within a period of 14 days, as they are more aware of these behaviors as a result of filling in the questionnaire. This awareness might induce subsequent behavior change, which is called the “question behavior effect” [42,43]. This can potentially explain the poor reliability scores, as participants think about their lifestyle behaviors and change their minds and feelings after filling in the questionnaire for the first time [42,43]. Moreover, a meta-analysis by Wilding et al. (2016) showed stronger “Question Behavior Effects” for the promotion of protective behaviors than for reducing risk behaviors [42]. This could be a possible explanation why questions regarding PA showed a higher number of low-reliability items compared to questions regarding SB or sleep. Questions on PA can be interpreted as promoting protective behavior (i.e., promotion of being active), whereas questions on SB or sleep can be interpreted as reducing risky behavior (i.e., reducing inactive behavior, reducing inconsistency in sleeping patterns).

Almost all explanatory constructs showed a good homogeneity. This means that items related to a specific construct can be combined and summed up in a single construct score. For some constructs, an insufficient Cronbach’s Alpha Coefficient was found, which indicates that caution should be taken when combining the items into one overall construct (i.e., autonomous motivation regarding PA, social modeling regarding PA, social support regarding PA, attitude regarding long sitting periods, internal behavioral control regarding sedentary behavior, self-efficacy regarding sitting time during household tasks, social support regarding limiting sedentary time during household tasks, social support regarding sleep, electronic devices and sleep environment in the home environment). A possible solution for dealing with these lower homogeneity scores is to delete specific items within each construct to create a better homogeneity among items [40]. However, deleting items within constructs did not improve the homogeneity scores, so none of the items were deleted. Another possibility is to combine subconstructs within the overall construct. For example, self-efficacy regarding limiting SB has different subconstructs for each active living domain, i.e., work, household, leisure time, and transport. When combining all items into the overarching “self-efficacy regarding limiting SB construct” there is sufficient homogeneity between items. Nevertheless, this will create a loss in detailed information on active living domains [40]. In order to not lose this information, every question can be interpreted separately, except for the items with poor reliability, which should not be used. For example, the “internal behavior control” construct regarding SB had a low homogeneity among items (α = 0.398). Deleting items within this construct did not improve the homogeneity of the construct. Therefore, it is recommended to not combine the items in an overall “internal behavior control” construct regarding SB, but to only use the items with moderate, good or excellent reliability as separate indicators of “internal behavior control”. It is suggested to do further research on constructs where the majority of items had a poor reliability. A recommendation could be to set up focus groups to test if these items and constructs are understandable and correctly interpreted by the target group.

One might question whether it is necessary to develop another questionnaire, since some questionnaires already exist that assess explanatory variables of lifestyle behaviors in isolation [12,15]. There are questionnaires to measure the explanatory variables of PA and SB, such as the DPAQ, and the Sedentary Behavior Change Questionnaire [12,15]. The DPAQ is a questionnaire with a good discriminant validity, test–retest reliability and a reasonable to good internal consistency for most determinant areas [12]. This questionnaire is based on the theoretical domains framework (TDF) which is a practical guide to identifying determinants to explain current behaviors [12,44,45]. Nevertheless, this theoretical framework is a pragmatic framework and does not rely upon a behavior change theory where the relation between determinant constructs and behavior intention is lacking [12,46]. However, using behavior change techniques out of the TDF as mediators to translate the behavior change theory into practice seems promising for promoting motivation for behavior change [23,47]. The Sedentary Behavior Change Questionnaire is a questionnaire (2019) based on the constructs of the SCT (i.e., self-efficacy, outcome expectations, goals setting, planning, and barriers) [15]. Moreover, this questionnaire is not optimally transferrable among other groups, as it was developed within a group of adults with multiple sclerosis [15]. This questionnaire reported some preliminary support for structural validity and internal consistency [15]. Additionally, there are different questionnaires on explanatory variables of sleep such as the Sleep Practices and Attitudes Questionnaire, or the Dysfunctional Beliefs and Attitudes about Sleep Scale (specific for insomnia), and the Sleep Beliefs Scale [48,49,50]. Most of these questionnaires investigate explanatory variables, which often lacks the link with a behavior change theory [48,49,50]. Moreover, none of these questionnaires addressed multiple behaviors such as 24-h movement behaviors, whereas this newly developed questionnaire does.

As the evidence on 24-h movement behavior paradigm is rapidly expanding, new challenges exist regarding the inclusion of this paradigm into health promotion research. Therefore, the main strength of this study is that this is the first questionnaire assessing the explanatory variables of all behaviors from the perspective of an entire day based on a theoretical framework, i.e., the socio-ecological model in combination of the IBC model embedded within the intrapersonal level. The IBC model is a behavior change model that integrates different psychological constructs from different behavior change theories, i.e., TPB, SCT, Dual System theory, and SDT [35]. The strength of using this theoretical framework is the ability to focus on the most important key constructs for behavior change [35]. Moreover, it showed a good fit with predicting PA behavior within a group of adults and older adults [20]. The key constructs of the TPB, SCT, SDT were mostly positively correlated with behavior intention [22,23,47]. However, in these behavior change theories, the behavior change intentions were often not sufficient to prompt effective changes, which can be explained by the weak links between key constructs or a disrupted link between intention and behavior change, also known as the intention behavior gap [35,51,52]. To bridge this gap, the IBC model integrated some constructs from the previously mentioned behavior change theories (TPB, SDT, SCT) with constructs of the Dual System theory (motivational phase, volitional phase) [20,35,52]. Additionally, as this newly developed questionnaire is a comprehensive assessment of different explanatory variables for all 24-h movement behaviors, it is possible to select a specific behavior of interest in combination with the construct of interest (e.g., self-efficacy regarding PA). However, it remains recommended to use the questionnaire in its entirety, as it provides the most valuable insights into behaviors across a total day. Additionally, this questionnaire has its fundaments in a behavior change theory that can lay the foundation for personalizing interventions in the future [53].

This study also has some limitations. First, this study used a convenience sampling method within the researchers’ network. Moreover, one of the inclusion criteria was formulated as working for at least 50 percent, because behaviors at work were addressed by questions in this questionnaire. This might not be representative for lower SES groups. Further studies are required to test this questionnaire within a group with lower SES. Second, this questionnaire was developed in Dutch and conducted in Flanders (Belgium). There is an English translation of questions available in the Appendix A; however, this English translation has not been tested. Third, this study was conducted within a general healthy adult population. Although, it is possible to add and test disease-specific items to each construct. For example, a question addressing barriers to performing PA among participants with diabetes may take a form such as “controlling my glucose level is a barrier to performing physical activity”. Last, this questionnaire included some poor-reliability questions as well as constructs with low homogeneity among items. It is recommended to use these questions and constructs with caution. Appendix A provides a detailed description of each question with the corresponding ICC, and the lower and upper bound. This creates the opportunity to use questions or constructs independently of the total questionnaire, as well as to adjust or further develop poor reliable questions and constructs. An additional recommendation for future research is to combine the assessment of this questionnaire with an objective assessment of 24-h movement behaviors (e.g., using an accelerometer). By collecting adults’ 24-h movement behaviors, these behaviors can be associated with the explanatory variables to provide a detailed assessment of lifestyle behaviors and to provide direction for the development of tailored interventions. Combining both objective and subjective assessment methods will provide the most complete information as both methods measure different aspects of behaviors (i.e., objectively measured behaviors, context, and explanatory variables).

## 5. Conclusions

This newly developed questionnaire on explanatory variables for 24-h movement behavior showed a moderate to excellent test–retest reliability for a majority of items and a good homogeneity among items for a majority of constructs. This comprehensive questionnaire, which collects a broad range of explanatory variables of all behaviors performed in a 24-h day, might be a valuable tool for future research to improve the understanding of adults’ 24-h movement behaviors. Hence, constructs with poor-reliability questions should be interpreted with caution, and it is recommended that, for constructs with an insufficient homogeneity among items, the items should be interpreted separately. Further research is recommended to fine tune poor-reliability questions or insufficiently homogenous constructs, as well as to adapt and test this questionnaire for use in clinical populations.

## Figures and Tables

**Figure 1 ijerph-20-04407-f001:**
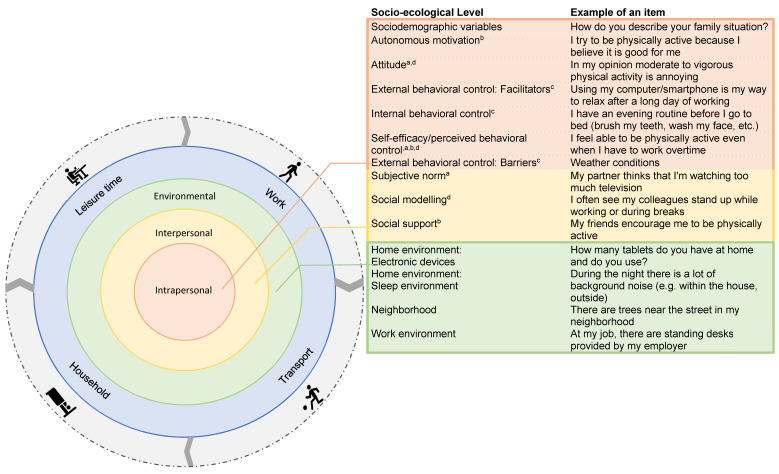
Questionnaire items within the socio-ecological model. The orange zone represents the intrapersonal level. The yellow zone represents the interpersonal level. The green zone represents the environmental level. The blue zone represents the four active living domains. The grey zone represents the 24-h movement behaviors including sleep, sedentary behavior, light physical activity and moderate to vigorous physical activity. The psychological factors questioned in the intra- and interpersonal level are part of a behavior change theory: Theory of Planned Behavior ^a^, Self-Determination Theory ^b^, Dual System Theory ^c^, Social Cognitive Theory ^d^.

**Table 1 ijerph-20-04407-t001:** Sample characteristics.

Sample Characteristics	Descriptive Numbers
Total sample size recruited (#)	40
Drop out (#)	5
Total sample size participated (#)	35
Age in years (mean (SD))	42.94 (±16.07)
Sex: female (# (%))	21 (60.00)
Unemployed (# (%))	3 (8.57)
Early retired/retired (# (%))	4 (11.43)
High educational level (# (%))	23 (65.70)
Net family income >2000 euro/month (# (%))	28 (80.00)
Average time between T1–T2 (mean in days (SD))	15.82 (±2.00)

#: number of participants, SD: standard deviation, T1: timepoint 1, T2: timepoint 2.

**Table 2 ijerph-20-04407-t002:** Summary of test–retest reliability questionnaire items and internal consistency of explanatory variable constructs.

Explanatory Variable Constructs	Items	Test–Retest Reliability (ICC)					IC (α)
			Excellent	Good	Moderate	Poor	
	n	ICC-Range	n (%)	n (%)	n (%)	n (%)	α
General information							
Intrapersonal level							
Sociodemographic variables	14	0.758–1.000	13 (92.86)	1 (7.14)	0	0	NA
Physical activity							
Intrapersonal level							
Autonomous motivation	2	0.322–0.577	0	0	1 (50.00)	1 (50.00)	*0.585*
Attitude: Overall	10	0.071–0.616	0	0	3 (30.00)	7 (70.00)	0.840
*Attitude LPA*	5	0.193–0.616	0	0	1 (20.00)	4 (80.00)	0.861
*Attitude MVPA*	5	0.071–0.571	0	0	2 (40.00)	3 (60.00)	0.811
Facilitators	14	0.141–0.654	0	0	3 (21.43)	11 (78.57)	0.769
Internal behavioral control	2	0.344–0.781	0	1 (50.00)	0	1 (50.00)	0.772
Self-efficacy	10	0.443–0.763	0	1 (10.00)	7 (70.00)	2 (20.00)	0.896
Barriers	16	0.054–0.821	0	3 (18.75)	8 (50.00)	5 (31.25)	0.942
Interpersonal level							
Subjective norm	5	0.416–0.851	0	1 (20.00)	3 (60.00)	1 (20.00)	0.869
Social modeling	5	0.693–0.849	0	4 (80.00)	1 (20.00)	0	*0.561*
Social support	6	0.071–0.826	0	2 (33.33)	2 (33.33)	2 (33.33)	*0.623*
**Summary physical activity**	**70**	**0.054–0.851**	**0**	**12 (17.14)**	**28 (40.00)**	**30 (42.86)**	
Sedentary behavior							
Intrapersonal level							
Autonomous motivation	2	0.218–0.554	0	0	1 (50.00)	1 (50.00)	0.789
Attitude: Long sitting period	5	0.106–0.720	0	0	3 (60.00)	2 (40.00)	*0.695*
Attitude: Interrupting sitting period	5	0.159–0.724	0	0	2 (40.00)	3 (60.00)	0.804
Facilitators: Overall	8	0.485–0.846	0	2 (25.00)	5 (62.50)	1 (12.50)	0.788
*Leisure time*	4	0.485–0.766	0	1 (25.00)	2 (50.00)	1 (25.00)	0.722
*Work*	2	0.561 −0.744	0	0	2 (100.00)	0	0.876
*Household*	2	0.670–0.846	0	1(50)	1 (50.00)	0	0.910
Internal behavioral control	5	0.089–0.782	0	1 (20.00)	2 (40.00)	2 (40.00)	*0.398*
Self-efficacy: Overall	17	0.065–0.848	0	1 (5.88)	8 (47.06)	8 (47.06)	0.855
*Leisure time*	7	0.222–0.740	0	0	3 (42.86)	4 (57.14)	0.744
*Transport*	4	0.425–0.654	0	0	2 (50.00)	2 (50.00)	0.729
*Work*	3	0.550–0.848	0	1 (33.33)	2 (66.67)	0	0.835
*Household*	3	0.065–0.677	0	0	1 (33.33)	2 (66.67)	*0.533*
Barriers: Overall	16	0.024–0.781	0	1 (6.25)	8 (50.00)	7 (43.75)	0.874
*Leisure time*	4	0.227–0.436	0	0	0	4 (100.00)	0.812
*Transport*	5	0.491–0.781	0	1 (20.00)	3 (60.00)	1 (20.00)	0.801
*Work*	5	0.594–0.696	0	0	5 (100.00)	0	0.951
*Household*	2	0.024–0.239	0	0	0	2 (100.00)	0.809
Interpersonal level							
Subjective norm: Overall	12	0.435–0.860	0	3 (25.00)	5 (41.67)	4 (33.33)	0.887
*Leisure time*	6	0.470–0.826	0	2 (33.33)	3 (50.00)	1 (16.67)	0.856
*Transport*	3	0.435–0.860	0	1 (33.33)	0	2 (66.67)	*0.638*
*Work*	1	0.635	0	0	1 (100.00)	0	NA
*Household*	2	0.498–0.730	0	0	1 (50.00)	1 (50.00)	0.732
Social modeling: Overall	20	0.377–0.911	3 (15.00)	7 (35.00)	8 (40.00)	2 (10.00)	0.724
*Leisure time*	12	0.588–0.911	3 (25.00)	5 (41.57)	4 (33.33)	0	0.771
*Transport*	4	0.553–0.853	0	2 (50.00)	2 (50.00)	0	0.814
*Work*	2	0.584–0.655	0	0	2 (100.00)	0	0.874
*Household*	2	0.377–0.428	0	0	0	2 (100.00)	0.779
Social support: Overall	12	0.498–0.787	0	3 (25.00)	8 (66.67)	1 (8.33)	0.910
*Leisure time*	6	0.530–0.779	0	2 (33.33)	4 (66.67)	0	0.886
*Transport*	3	0.502–0.787	0	1 (33.33)	2 (66.67)	0	0.789
*Work*	1	0.612	0	0	1 (100.00)	0	NA
*Household*	2	0.498–0.714	0	0	1 (50.00)	1 (50.00)	0.645
**Summary sedentary behavior**	**102**	**0.024–0.911**	**3 (2.94)**	**18 (17.65)**	**50 (49.02)**	**31 (30.39)**	
Sleep							
Intrapersonal level							
Autonomous motivation	2	0.260–0.379	0	0	0	2 (100.00)	0.817
Attitude: Overall	10	0.079–0.724	0	0	5 (50.00)	5 (50.00)	0.742
*Optimal sleep pattern*	5	0.079–0.605	0	0	2 (40.00)	3 (60.00)	0.734
*Electronic devices*	5	0.407–0.724	0	0	3 (60.00)	2 (40.00)	0.749
Facilitators	6	0.315–0.640	0	0	3 (50.00)	3 (50.00)	0.867
Internal behavioral control	2	0.630–0.641	0	0	2 (100.00)	0	0.760
Self-efficacy	6	0.461–0.812	0	1 (16.67)	3 (50.00)	2 (33.33)	0.804
Barriers: Overall	12	0.156–0.730	0	1 (8.33)	9 (75.00)	2 (16.67)	0.821
*Optimal sleep pattern*	9	0.156–0.730	0	0	7 (77.78)	2 (22.22)	0.807
*Electronic devices*	3	0.578–0.848	0	1 (33.33)	2 (66.67)	0	0.748
Interpersonal level							
Subjective norm	2	0.522–0.656	0	0	2 (100)	0	0.745
Social modeling	3	0.679–0.805	0	2 (66.67)	1 (33.33)	0	0.832
Social support	2	0.446–0.526	0	0	1 (50.00)	1 (50.00)	*0.687*
**Summary sleep**	**45**	**0.079–0.848**	**0**	**4 (8.89)**	**26 (57.78)**	**15 (33.33)**	
Environment							
Physical environment level							
Home environment: Electronic devices	10	0.622–0.896	0	7 (70.00)	3 (30.00)	0	*0.664*
Home environment: Sleep environment	7	0.474–0.850	0	5 (71.43)	1 (14.29)	1 (14.29)	*0.526*
Neighborhood	13	0.437–0.934	2 (15.38)	3 (23.08)	7 (53.85)	1 (7.69)	0.797
Work environment	5	0.823–1.000	3 (60.00)	2 (40.00)	0	0	0.916

ICC: Intraclass correlation coefficient with cut-off points for Excellent (ICC ≥ 0.900), Good (ICC 0.750–0.899), Moderate (ICC 0.50–0.749), and Poor (ICC < 0.500) reliability; IC: internal consistency, indicated by Cronbach’s Alpha Coefficient (α). Italic α (α < 0.70) represents poor homogeneity among items. Number of items is represented as the number of items (n) and the percentage (%) of the corresponding construct; **Bold explanatory variable constructs**: summary of all items related to a 24-h movement behaviors; *Italic explanatory variable constructs*: Subconstructs where the number of items are combined in the overall construct; LPA: light physical activity; MVPA: moderate to vigorous physical activity; NA: not applicable.

## Data Availability

Data sharing is applicable upon request.

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
