# Peer review of "Test–Retest Reliability and Internal Consistency of a Newly Developed Questionnaire to Assess Explanatory Variables of 24-h Movement Behaviors in Adults"

_ijerph, 2023, doi:10.3390/ijerph20054407_

Round 1

Reviewer 1 Report

This is an excellent paper that aimed to examine the test-retest reliability and internal consistency of a newly developed questionnaire on 88 explanatory variables of 24-hour movement behaviors among adults. Here are some comments that I would like the authors to address.

1. In the 2nd paragraph, please provide more arguments that the authors use to justify the importance of 24-hour movement behavior. How does it compare to the other movement behaviors such as one-week physical activity?

2. The reviewer thinks that the sample size is relatively small. Please provide a detailed justification for selecting 30 adults for the sample size? (line 93).

3. “…40 adults with a minimum age of 18 years old were recruited in Flanders…” (line 95). The mean age of participants was 42.9±16.07. Why the authors selected participants who are in the same age group? Did not the authors think that the broad age range affects the study results?

4. Please replace some out-of-date references with new ones. 

Reviewer 2 Report

MAJOR COMMENTS

The present study investigated the test-retest reliability and internal consistency of a newly developed questionnaire on explanatory variables of 24-hour movement behaviors among adults based on a socio-ecological model. The work is relevant, and questionnaires of this nature contribute to the collection of less expensive information.

In general, throughout the entire text, the authors include sleep as part of the 24-hour movement behavior. It must be considered that, by definition, sleep is a functional, reversible, and cyclic state whose one of its main characteristics is the absence or abrupt reduction of movement. The presence of movements during sleep is typified as sleep-related movement disorders. Therefore, although the term 24-hour movement behaviors are widely used in the specific literature, it is a misleading term from the point of view of chronobiology or sleep science. Still, I would suggest that the term be adjusted for circadian behaviors.

In the introduction, there is a small mention of sleep, however, there is a lack of argumentative bases on this behavior, especially considering that sleep is a behavioral state that occupies, on average, 1/3 of the time in the 24 hours of a subject's day. I suggest that the relationship between sleep, circadian rhythm, and interpersonal and socioecological variables be better grounded in the introduction to the text.

In line 35, the authors state that doing physical activity and reducing physical inactivity are the main characteristics, however, contextually, it is necessary to include adequate sleep as one of the parameters of a healthy lifestyle. Poor sleep is also associated with impaired glucose metabolism, obesity, and cardiovascular problems.

MINOR COMMENTS

In lines 263 and 265 the authors state that they showed a moderate to excellent test-retest reliability of items of different constructs. How is it possible to define this rating from moderato to excellent? With how many variables with moderate or excellent retest reliability is it possible to determine this classification?

What was the average response time to the questionnaire in this study?

In line 283 the authors state that: “PA can be seen as a promoting protective behavior where SB and sleeping can be seen as risk behaviors” but do not explain why these behaviors can be interpreted in this way.

Between lines 312-333, the authors suggest that, although there are other questionnaires, the application of these questionnaires would not be feasible since his theoretical framework is a pragmatic framework and does not rely upon a behavior change theory. How to solve the problem of low reliability of answers for some constructs of this instrument? This was not made clear in the discussion.

The conclusion of this work must be assertive. Pointing out what are, in fact, the necessary precautions for issues of low reliability or homogeneity.
